# Inundation Analysis of Coastal Urban Area under Climate Change Scenarios

**Heechan Han** [1] , **Deokhwan Kim** [2] **and Hung Soo Kim** [3,*]

1   Blackland Research and Extension Center, Texas A&M AgriLife, Temple, TX 76502, USA; heechan.han@ag.tamu.edu
2   Department of Hydro Science and Engineering Research, Korea Institute of Civil Engineering and Building Technology (KICT), Goyang 10223, Korea; kimdeokhwan@kict.re.kr
3   Department of Civil Engineering, Inha University, Incheon 22212, Korea
*   Correspondence: sookim@inha.ac.kr

**Abstract:** The inundation of urban areas has frequently occurred as a result of the localized heavy precipitation and flash floods in both South Korea and globally. Metropolitan areas with higher property value and population density than rural areas need practical strategies to reduce flood damage. Therefore, this study aims to perform an inundation analysis of coastal urban areas under a climate change scenario. Changwon city is one of the typical coastal metropolitan regions in South Korea. Severe flooding has occurred in this area caused by a combination of precipitation and sea-level rise enhanced by the typhoon, Sanba, in September 2012. At that time, daily precipitation was 65.5 mm, which is lower than the capable amount of rainfall of the drainage system. However, the river stage combined with the tidal wave caused by a typhoon and heavy precipitation exceeded the flood warning level. This study performed the flood inundation analysis for a part of Changwon city using the SWMM model, a two-dimensional urban flood analysis model. Furthermore, we considered the climate change scenarios to predict the potential flood damage that may occur in the future. As a result, as the future target period increases, both the flooding area and the inundation depth increase compared to the results of the inundation simulation according to the current precipitation and sea-level conditions. The inundation area increased by 2.6–16.2% compared to the current state, and the flooded depths would be higher than 1 m or more. We suggest a structural method to reduce inundation damages to consider extreme precipitation and tidal wave effects.

**Keywords:** coastal urban area; inundation analysis; sea-level rise; climate change scenarios; SWMM

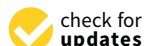

## 1. Introduction

Recently, frequent flash floods due to climate change and abnormal climate patterns have severely damaged the urban area. Specifically, unlike inland urban areas, coastal urban areas, which are affected by a combination of precipitation and sea-level patterns, have more severe damage even with the same amount of precipitation [1–4]. In coastal areas with high population density and high asset values, the damages caused by flooding are fatal and critical, so practical strategies are required to prevent flooding disasters in coastal areas. South Korea is vulnerable to flood disasters in coastal areas because many country regions are composed of the ocean [5,6]. Moreover, considering the sea-level rise and the extreme precipitation patterns due to climate change, it is necessary to analyze the vulnerability of flooding through a complex inundation simulation for coastal areas in South Korea.

In South Korea, the damage caused by typhoons in 2003 was 531 people (133 dead, 15 missing, 383 injured), and property damage was approximately 4.4 billion dollars. For example, Gyeongsangnam-do, which has many coastal urban areas in South Korea, suffered from fatal flooding damages in 2003. The flooding damage cost was approximately 2 trillion dollars, accounting for approximately 50% of the damage that occurred nationwide, and

flood damage was in an area of 27,000 ha. Therefore, it is essential to study the mitigation of flood damage by developing an early warning system in coastal areas vulnerable to rising sea levels and extreme precipitation patterns. For this, research on the flooding simulation model that can simulate and predict the inundation in urban areas considering the characteristics of precipitation and sea-level rise according to climate change should be accompanied first.

Research on urban inundation analysis caused by extreme precipitation and sea-level rise has been widely conducted worldwide, including in the US [7–9], Asia [10–14], Europe [15–17], and South Korea [5,18–20]. Specifically, inundation simulation and analysis have been performed using 1D and 2D numerical models and hydraulic models such as the Storm Water Management Model (SWMM) and MIKE models. For example, Hsu et al. (2000) [21] evaluated the rainwater treatment capacity of the sewer system and the pump station installed in Taipei by performing a flood simulation using the SWMM model. Hsu et al. (2002) [22] analyzed the correlation between the urban sewer system and the surface flow using a 2D non-inertia overland flow model and SWMM's EXTRAN module. In addition, Leandro et al. (2009) [23] analyzed the flood inundation in urban areas using a combination of the 1D sewer model and 2D surface network system. These studies have shown that physical-based models have sufficient power to simulate and analyze the inundation in urban areas.

In South Korea, where major cities are located along the coast, many studies on inundation analysis using urban flooding simulation models for coastal cities have been conducted [24–26]. For example, Jeong et al. (2011) [27] evaluated the impact of extreme precipitation and sea-level patterns enhanced by typhoons on inundation in urban areas using a 2D numerical model. They suggested using a hydraulic structure such as installing a drainage system in urban areas and emphasized the importance of a flood analysis computational model to prevent flood damage in advance. Kim et al. (2009) [28] conducted a flood simulation by considering the dual-drainage method that considers the outflow and inflow from the manhole simultaneously and the effect of the tidal patterns on the coast. Moon et al. (2006) [29] used the MIKE21 model to simulate the flooding of nine typhoons that caused large tidal waves over the past ten years. They showed how coastal urban areas are vulnerable to sea levels rising when a storm is coming. In addition, Kang et al. (2013) [30] performed an urban floods simulation by applying radar precipitation to SWMM to predict flash floods accurately and rapidly. However, since most studies have mainly investigated flood damage based on past conditions, studies on potential flood damage in the future are lacking.

Climate change scenarios are data that evaluate and predict the scientific, technological, and socio-economic impacts of climate change due to complex factors such as global future social, economic, and human activities and the resulting greenhouse gas emissions. The Intergovernmental Panel on Climate Change (IPCC) has published a climate change assessment report every 5 to 6 years and has consistently reported scientific evidence for climate change and its effects. One of the main tasks of the IPCC is to develop potential future climate change scenarios according to greenhouse gas emissions and evaluate the climate change response strategies.

Many researchers have evaluated the impact of future climate patterns on inundation in urban areas by considering climate change scenarios reported by the IPCC [31–36]. Specifically, in the case of studies targeting coastal urban areas, inundation analysis has been performed according to complex factors caused by both the extreme precipitation patterns and the sea-level rise at the same time. For example, Wang et al. (2012) [1] studied the effects of climate change on urban inundation and complex factors combining sea-level rise, storm surge, and subsidence in Shanghai, China using MIKE21 software. They found that approximately 46% of areas in Shanghai would be flooded by 2100. They suggested the need for a study on the effects of complex factors such as precipitation and sea level on coastal urban areas. In addition, there are notable studies that conducted hydrological

impact assessments of climate change using the latest scenario data (i.e., AR6 of IPCC) reflecting the latest information released [37,38].

Changwon city, one of the typical coastal cities in South Korea, has suffered inundation damage when a typhoon has hit, but disaster-prevention strategies and policies are insufficient. In particular, the typhoon in 2012, named Sanba, caused severe damage to property and humans. At that time, the maximum hourly precipitation was 25 mm, which was lower than the designed precipitation rate of the sewer pipeline system. Still, the sea-level rise in the ocean exceeded the warning level, and flood damage occurred. This is one of the examples of why we need to consider both the precipitation patterns and sea-level rise for inundation analysis in coastal urban areas, unlike inland regions. Therefore, this study aims to simulate inundation in Changwon City, South Korea, considering extreme precipitation and sea-level rise and predicting the potential vulnerability to flooding damage enhanced by climate change scenarios.

## 2. Materials and Methods

### 2.1. Study Area and Datasets

This study selected the Jangguncheon River Basin in Changwon city, South Korea (Figure 1). The basin area is approximately 6 km$^2$, and it is an urban area facing the ocean. The west side of the drainage basin consists of mountainous terrain. The east side is in contact with Masan Bay, so it has the characteristic that the runoff from precipitation flows out to the river and ocean through the drainage system under the surface. The area is frequently flooded with typhoons and torrential precipitation in summer and autumn every year. The average annual precipitation in the region is approximately 1500 mm/yr, and the amount of precipitation in the summer (Jun to Sep) accounts for approximately 60–70% of the annual precipitation.

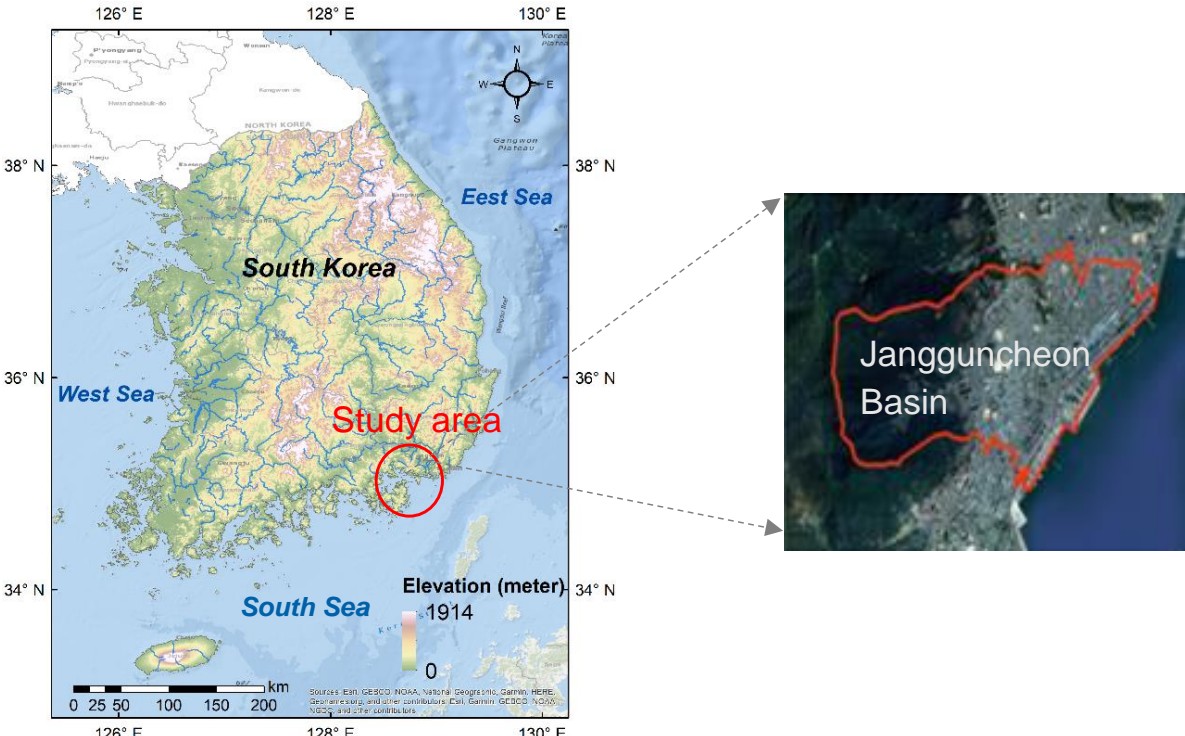

**Figure 1.** The application area of this study is Jangguncheon Basin in Changwon City, South Korea.

The datasets needed to simulate the urban floods using the SWMM model are two types, in terms of hydrological and hydraulic. The hydrological element includes precipitation events and topographical information data of the target area, and the hydraulic feature includes sewage pipeline network data. Observed precipitation or designed precipitation

data can be applied for precipitation events. Topographical information comprises the digital elevation model (DEM) and land cover features of the area. The sewage pipeline network data, including pipe features such as the height, shape, length, slope, and roughness coefficient, are used for the hydraulic dataset. In this study, precipitation data, the primary input data for the SWMM model, were collected by the Korea Meteorological Administration (KMA) [39]. The Korean Water Resources Management Information System (WAMIS) [40] and tide data were collected through the Korean Hydrographic and Oceanographic Agency (KHOA) [41]. In addition, the sewage pipeline network input data of SWMM, composed of 430 manholes and 467 conduits, were constructed using the pipeline network data of the blueprint, which are installed in the study area.

Figure 2 shows the flowchart of this study. We used designed precipitation as input data of the SWMM model. Specifically, the Huff 2nd curve [42] was applied to design precipitation rates with 10-, 30-, and 50-year return periods and 1–12 h durations. In addition, the predicted precipitation from 2010 to 2100 provided from the IPCC's Fourth Assessment Report (AR4) climate change scenario was also used as input data for the SWMM. Here, the future period was divided into three target periods, target periods I (2010–2040), II (2040–2070), and III (2070–2100), for practical inundation result analysis. In addition to the precipitation scenario, the sea-level data observed at the tide station in the basin and future sea-level rise scenarios are added to the model.

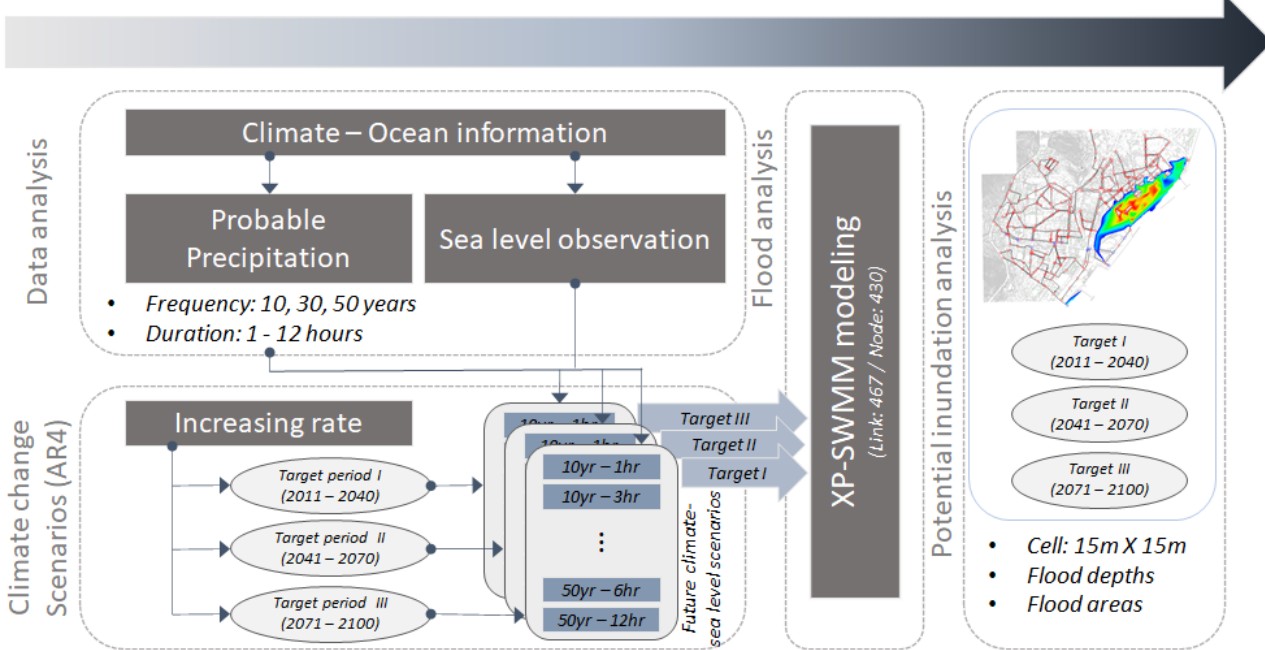

**Figure 2.** Flowchart of this study.

### 2.2. SWMM Model

The SWMM model is a comprehensive model that can simulate the surface and subsurface flow of runoff and pollutants caused by precipitation events within an urban watershed [43]. In this study, the XP-SWMM model was used for inundation analysis. XP-SWMM (hereinafter SWMM model) is windowed to provide visual functions so that users of the SWMM model can easily build and perform models for flood analysis. The basic analysis method and structure are the same as that of the traditional SWMM model. The XP-SWMM engine is based on the EXTRAN, TRANSPORT, and STORAGE/TREATMENT modules of the US EPA SWMM (version 4) engine [44,45]. The SWMM model is stormwater network analysis software based on link and node characteristics. Thus, it is necessary to build datasets including sewer pipeline networks in the area and specifications and

attributes for each link and node. The SWMM model consists of four computational and five service blocks, as shown in Figure 3.

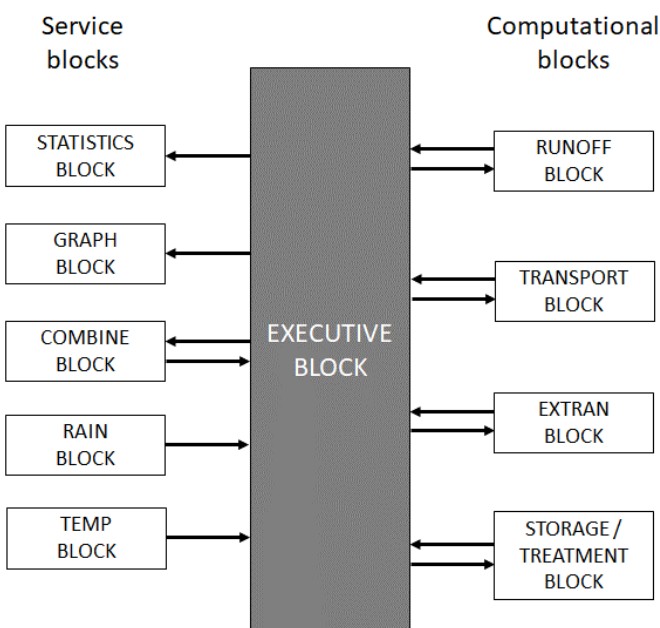

**Figure 3.** Structure of SWMM model.

　　The primary role of the RUNOFF block is to simulate runoff and water quality changes in the drainage basin for precipitation events and performs the initial calculation of the SWMM model. In addition, the surface and subsurface runoff are simulated using data such as the precipitation pattern, antecedent precipitation condition, land use, and topography. The TRANSPORT block uses the simulated result data from the RUNOFF block to track the water flow and pollutants in the drainage system during precipitation events and simulates the infiltration process into the drainage system. The EXTRAN block aims to calculate the water flow and depth in the drainage system, and hydrodynamic equations affect the flow in channels. In addition, the STORAGE block tracks water flow and contaminants passing through storage and treatment facilities.

　　In this study, as hydraulic input data for driving the SWMM model, a total of 430 manholes (nodes) that deliver precipitation that falls onto the surface of the sewer pipeline were used (Figure 4). Furthermore, 467 channels (links) connecting two nodes were used. Different features of each link, including the shape, diameter, length, pipe height, the slope of the pipe, and the roughness coefficient, were considered in this study. Assuming that it is composed of concrete, a roughness coefficient of 0.014 was applied equally to all pipelines. Before simulating flood damage using the SWMM model it is necessary to verify whether the model can adequately simulate the inundation features of the target watershed. Due to the absence of actual observation data for model verification, there is a limitation in that the model was verified using only the historical flood map in the area. The watershed has a history of flooding areas caused by Typhoon "Sanba" on 16–17 September 2012. Therefore, in this study, flood simulation was performed by applying the actual precipitation and sea-level data that occurred on 16–17 September 2012 and comparing them with the actual flooding area at that time. The simulation results and the flooded area that occurred at that time were found to be approximately 90% consistent, and the verified model was applied to the simulation of potential flood damage in the future.

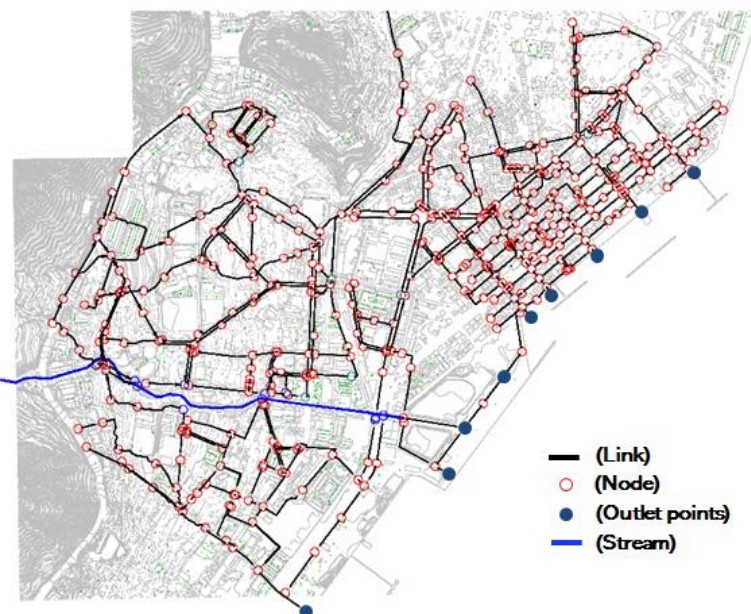

**Figure 4.** Hydraulic input data (i.e., link, node, and outlet points) of SWMM in Changwon City.

### 2.3. Climate Change Scenario

In the IPCC 4th Assessment Report (AR4 scenario), greenhouse gas emission trends (i.e., A1B, B2, and A2) were considered to calculate future climate change patterns such as precipitation and temperature worldwide. Each emission scenario was projected considering various conditions (e.g., economy, population, energy, etc.). According to the IPCC, none of the AR4 scenarios represent a "best guess" of future emissions because any of the AR4 scenarios were more likely to occur than others. In addition, high-resolution climate change scenarios, including precipitation, temperatures, and sea-level patterns in South Korea, were calculated using detailed statistical techniques and global climate models (GCMs).

The climate change scenario presents predicted weather patterns up to 2100 for global climate characteristics. For example, according to the B1 and A2 scenarios, by 2100, the global average temperature will increase by approximately 1.8 °C and 3.4 °C, and sea-level rise likely ranges from 18 to 38 cm (for the B1 scenario) and from 23 to 51 cm (for the A2 scenario) [46]. In addition, precipitation will also increase between the wet and dry seasons. It was predicted that the increasing rate would be more significant, and the average temperature and sea-level rise were also predicted to be significantly increased in the future. In Changwon City, the application area of this study, the annual precipitation increased by approximately 25 mm every ten years, and annual precipitation will have reached 1800 mm in 2100, which increased by over 20% compared to the present, and the temperature was predicted to increase by 30% [47].

In this study, based on the AR4 scenario (i.e., A1B and B1), precipitation and sea-level changes from 2010 to 2100 were applied to the inundation simulation to analyze the impact of climate change on the future inundation of the Changwon region in South Korea. The future precipitation and sea-level rise data are based on information provided by the Korean National Emergency Management Agency (2011) [47]. For more efficient analysis, the scenarios were divided into three phases of time, and inundation simulations were performed for the initial (target period I, 2010–2040), mid-term (target period II, 2040–2070), and long-term (target period III, 2070–2100) periods.

### 2.4. Potential Probable Precipitation with Climate Change Scenario

In this study, precipitation designed with a frequency of 10 years or more, the design standard for trunk sewer pipes, was applied. To simulate various precipitation scenarios,

twelve scenarios were considered in this study. Each scenario includes probable precipitation events for short-term (10 years), medium-term (30 years), and long-term (50 years) frequency and duration (1 to 12 h) for each frequency duration. A duration of 1 to 12 h was adopted to consider the characteristics of an urban area with large impervious areas and small basin areas. Table 1 represents the probable precipitation by frequency duration in Changwon City, South Korea.

**Table 1.** The probable precipitation in Changwon City, South Korea. A grey box indicates the probable precipitation values used in this study [47].

| Frequency (Year) | Duration (Hour) | | | |
|---|---|---|---|---|
| | 1 | 3 | 6 | 12 |
| 2 | 39.6 | 73.5 | 104.6 | 139.1 |
| 3 | 42.5 | 79.0 | 111.5 | 147.8 |
| 5 | 45.7 | 85.0 | 119.2 | 157.4 |
| **10** | **49.7** | **92.7** | **128.8** | **169.6** |
| **30** | **55.8** | **104.2** | **143.4** | **188.0** |
| **50** | **58.5** | **109.5** | **150.1** | **196.4** |
| 70 | 60.4 | 112.9 | 154.4 | 201.9 |
| 80 | 61.1 | 114.3 | 156.2 | 204.0 |
| 100 | 62.3 | 116.6 | 159.0 | 207.7 |
| 200 | 66.0 | 123.7 | 168.0 | 218.9 |
| 500 | 70.9 | 133.0 | 179.8 | 233.8 |

This study distributed the probability precipitation of six scenarios for each duration using the Huff method [42]. The Huff method is one of the time distribution methods of precipitation to determine the design precipitation event required to estimate the design flood volume for hydrological structures. The method divides the precipitation duration into four equal sections and determines the time distribution characteristics of precipitation into four types according to which section the most significant precipitation occurs. In this study, the Huff 2nd method was applied to distribute the 10-, 30-, and 50-year probable precipitation for durations of 1 to 12 h in Changwon City.

Table 2 represents the designed probable precipitation in Changwon City, calculated by the precipitation with a frequency of 10, 30, and 50 years and a duration of 1 to 12 h to the Huff 2nd method. The precipitation designed according to the Huff 2nd distribution was determined so that the maximum precipitation occurred 24 min after the occurrence of precipitation for a duration of 1 h, 72 min for a duration of 3 h, 144 min for a duration of 6 h, and 360 min for a duration of 12 h.

**Table 2.** Designed probable precipitation with the frequency of 10, 30, and 50 years and duration of 1 h (60 min) to 12 h (720 min).

| Duration (min) | Probable Precipitation (mm) | | | Duration (min) | Probable Precipitation (mm) | | |
|---|---|---|---|---|---|---|---|
| | 10 Year | 30 Year | 50 Year | | 10 Year | 30 Year | 50 Year |
| 0 | 0.00 | 0.00 | 0.00 | 0 | 0.00 | 0.00 | 0.00 |
| 6 | 2.12 | 2.38 | 2.49 | 18 | 3.95 | 4.44 | 4.66 |
| 12 | 4.20 | 4.71 | 4.95 | 36 | 7.83 | 8.81 | 9.26 |
| 18 | 8.42 | 9.46 | 9.91 | 54 | 15.72 | 17.66 | 18.56 |
| 24 | 10.45 | 11.74 | 12.30 | 72 | 19.49 | 21.91 | 23.03 |
| 30 | 9.37 | 10.52 | 11.03 | 90 | 17.48 | 19.64 | 20.64 |

**Table 2.** *Cont*.

| Duration (min) | Probable Precipitation (mm) | | | Duration (min) | Probable Precipitation (mm) | | |
|---|---|---|---|---|---|---|---|
| | 10 Year | 30 Year | 50 Year | | 10 Year | 30 Year | 50 Year |
| 36 | 6.38 | 7.15 | 7.51 | 108 | 11.89 | 13.37 | 14.04 |
| 42 | 3.47 | 3.90 | 4.09 | 126 | 6.48 | 7.29 | 7.66 |
| 48 | 2.21 | 2.49 | 2.60 | 144 | 4.12 | 4.63 | 4.87 |
| 54 | 2.6 | 2.65 | 2.78 | 162 | 4.40 | 4.95 | 5.20 |
| 60 | 0.72 | 0.80 | 0.84 | 180 | 1.34 | 1.50 | 1.58 |
| 0 | 0.00 | 0.00 | 0.00 | 0 | 0.00 | 0.00 | 0.00 |
| 36 | 5.48 | 6.11 | 6.39 | 72 | 7.22 | 8.00 | 8.36 |
| 72 | 10.89 | 12.12 | 12.69 | 144 | 14.34 | 15.90 | 16.61 |
| 108 | 21.84 | 24.31 | 25.45 | 216 | 28.75 | 31.87 | 33.29 |
| 144 | 27.08 | 30.15 | 31.56 | 288 | 25.66 | 39.53 | 41.30 |
| 180 | 24.28 | 27.03 | 28.29 | 360 | 31.97 | 35.44 | 37.02 |
| 216 | 16.52 | 18.40 | 19.26 | 432 | 21.76 | 24.11 | 25.19 |
| 252 | 9.01 | 10.02 | 10.49 | 504 | 11.86 | 13.15 | 13.74 |
| 288 | 5.73 | 6.39 | 6.68 | 576 | 7.54 | 8.37 | 8.74 |
| 324 | 6.11 | 6.80 | 7.13 | 648 | 8.06 | 8.92 | 9.32 |
| 360 | 1.86 | 2.07 | 2.16 | 720 | 2.44 | 2.71 | 2.83 |

In addition, this study used the probable precipitation for each target period (i.e., I, II, and III) for the duration of 1 to 12 h impacted by the AR4 climate change scenario by applying the increase rate (12.1–22.4% for a 1 h duration, 9.9–18.8% for a 3 h duration, 10.9–16.5% for a 6 h duration, and 6.2–8.7% for 12 h duration) in probable precipitation calculated using the regional frequency analysis method presented by the Korean National Emergency Management Agency (2011) [47]. Tables 3–6 represent the predicted probable precipitation with three frequencies (i.e., 10, 30, and 50 years) and four durations (1, 3, 6, and 12 h) for three target periods in Changwon City.

**Table 3.** Predicted probable precipitation for three target periods for the duration of 1 h in Changwon City [47].

| Duration (min) | 2011–2040 (Target I) | | | 2041–2070 (Target II) | | | 2071–2100 (Target III) | | |
|---|---|---|---|---|---|---|---|---|---|
| | Precipitation (mm) | | | Precipitation (mm) | | | Precipitation (mm) | | |
| | 10 Year | 30 Year | 50 Year | 10 Year | 30 Year | 50 Year | 10 Year | 30 Year | 50 Year |
| 0 | 0.00 | 0.00 | 0.00 | 0.00 | 0.00 | 0.00 | 0.00 | 0.00 | 0.00 |
| 6 | 2.39 | 2.67 | 2.79 | 2.39 | 2.67 | 2.79 | 2.57 | 2.90 | 3.05 |
| 12 | 4.74 | 5.29 | 5.55 | 4.74 | 5.29 | 5.55 | 5.09 | 5.75 | 6.05 |
| 18 | 9.51 | 10.61 | 11.11 | 9.51 | 10.61 | 11.11 | 10.22 | 11.54 | 12.14 |
| 24 | 11.80 | 13.17 | 13.79 | 11.80 | 13.17 | 13.79 | 12.68 | 14.32 | 15.06 |
| 30 | 10.5 | 11.80 | 12.37 | 10.5 | 11.80 | 12.37 | 11.36 | 12.83 | 13.50 |
| 36 | 7.20 | 8.03 | 8.41 | 7.20 | 8.03 | 8.41 | 7.74 | 8.73 | 9.18 |
| 42 | 3.92 | 4.38 | 4.58 | 3.92 | 4.38 | 4.58 | 4.21 | 4.76 | 5.01 |
| 48 | 2.50 | 2.78 | 2.92 | 2.50 | 2.78 | 2.92 | 2.69 | 3.03 | 3.18 |
| 54 | 2.66 | 2.98 | 3.11 | 2.66 | 2.98 | 3.11 | 2.86 | 3.24 | 3.40 |
| 60 | 0.81 | 0.90 | 0.95 | 0.81 | 0.90 | 0.95 | 0.87 | 0.98 | 1.03 |

**Table 4.** Predicted probable precipitation for three target periods for the duration of 3 h in Changwon City [47].

| Duration (min) | 2011–2040 (Target I) Precipitation (mm) | | | 2041–2070 (Target II) Precipitation (mm) | | | 2071–2100 (Target III) Precipitation (mm) | | |
|---|---|---|---|---|---|---|---|---|---|
| | 10 Year | 30 Year | 50 Year | 10 Year | 30 Year | 50 Year | 10 Year | 30 Year | 50 Year |
| 0 | 0.00 | 0.00 | 0.00 | 0.00 | 0.00 | 0.00 | 0.00 | 0.00 | 0.00 |
| 18 | 4.42 | 4.90 | 5.12 | 4.42 | 4.90 | 5.12 | 4.69 | 5.11 | 5.24 |
| 36 | 8.77 | 9.74 | 10.18 | 8.77 | 9.74 | 10.18 | 9.31 | 10.15 | 10.41 |
| 54 | 17.58 | 19.52 | 20.40 | 17.58 | 19.52 | 20.40 | 18.67 | 20.35 | 20.86 |
| 72 | 21.81 | 24.21 | 25.30 | 21.81 | 24.21 | 25.30 | 23.16 | 25.24 | 25.88 |
| 90 | 19.56 | 21.70 | 22.69 | 19.56 | 21.70 | 22.69 | 20.76 | 22.63 | 23.20 |
| 108 | 13.30 | 14.77 | 15.43 | 13.30 | 14.77 | 15.43 | 14.12 | 15.39 | 15.79 |
| 126 | 7.26 | 8.05 | 8.42 | 7.26 | 8.05 | 8.42 | 7.70 | 8.40 | 8.60 |
| 144 | 4.61 | 5.13 | 5.35 | 4.61 | 5.13 | 5.35 | 4.90 | 5.34 | 5.48 |
| 162 | 4.93 | 5.46 | 5.72 | 4.93 | 5.46 | 5.72 | 5.23 | 5.70 | 5.85 |
| 180 | 1.49 | 1.66 | 1.73 | 1.49 | 1.66 | 1.73 | 1.59 | 1.73 | 1.77 |

**Table 5.** Predicted probable precipitation for three target periods for the duration of 6 h in Changwon City [47].

| Duration (min) | 2011–2040 (Target I) Precipitation (mm) | | | 2041–2070 (Target II) Precipitation (mm) | | | 2071–2100 (Target III) Precipitation (mm) | | |
|---|---|---|---|---|---|---|---|---|---|
| | 10 Year | 30 Year | 50 Year | 10 Year | 30 Year | 50 Year | 10 Year | 30 Year | 50 Year |
| 0 | 0.00 | 0.00 | 0.00 | 0.00 | 0.00 | 0.00 | 0.00 | 0.00 | 0.00 |
| 36 | 6.11 | 6.77 | 7.07 | 6.19 | 6.77 | 7.07 | 6.39 | 6.91 | 7.15 |
| 72 | 12.13 | 13.45 | 14.05 | 12.28 | 13.45 | 14.05 | 12.69 | 13.73 | 14.20 |
| 108 | 24.32 | 26.95 | 28.17 | 24.63 | 26.95 | 28.17 | 25.43 | 27.51 | 29.47 |
| 144 | 30.17 | 33.45 | 34.94 | 30.55 | 33.45 | 34.94 | 31.55 | 34.14 | 35.32 |
| 180 | 27.05 | 29.97 | 31.32 | 27.39 | 29.97 | 31.32 | 28.29 | 30.60 | 31.67 |
| 216 | 18.40 | 20.40 | 21.31 | 18.63 | 20.40 | 21.31 | 19.25 | 20.82 | 21.54 |
| 252 | 10.04 | 11.12 | 11.62 | 10.16 | 11.12 | 11.62 | 10.49 | 11.35 | 11.74 |
| 288 | 6.38 | 7.08 | 7.40 | 6.47 | 7.08 | 7.40 | 6.68 | 7.22 | 7.48 |
| 324 | 6.84 | 7.55 | 7.89 | 6.89 | 7.55 | 7.89 | 7.12 | 7.71 | 7.97 |
| 360 | 2.06 | 2.29 | 2.39 | 2.10 | 2.29 | 2.39 | 2.16 | 2.34 | 2.42 |

**Table 6.** Predicted probable precipitation for three target periods for the duration of 12 h in Changwon City [47].

| Duration (min) | 2011–2040 (Target I) Precipitation (mm) | | | 2041–2070 (Target II) Precipitation (mm) | | | 2071–2100 (Target III) Precipitation (mm) | | |
|---|---|---|---|---|---|---|---|---|---|
| | 10 Year | 30 Year | 50 Year | 10 Year | 30 Year | 50 Year | 10 Year | 30 Year | 50 Year |
| 0 | 0.00 | 0.00 | 0.00 | 0.00 | 0.00 | 0.00 | 0.00 | 0.00 | 0.00 |
| 72 | 7.77 | 8.50 | 8.84 | 7.78 | 8.51 | 8.84 | 7.85 | 8.55 | 8.84 |
| 144 | 15.43 | 16.88 | 17.55 | 15.46 | 16.90 | 17.55 | 15.59 | 16.97 | 17.55 |
| 216 | 30.93 | 33.85 | 35.19 | 30.99 | 33.87 | 35.19 | 31.25 | 34.04 | 35.19 |
| 288 | 38.38 | 41.98 | 43.65 | 38.45 | 42.02 | 43.65 | 38.76 | 42.22 | 43.65 |
| 360 | 34.40 | 37.64 | 39.14 | 34.46 | 37.68 | 39.14 | 34.76 | 37.85 | 39.14 |
| 432 | 23.40 | 25.60 | 26.62 | 23.45 | 25.63 | 26.62 | 23.64 | 25.75 | 26.62 |
| 504 | 12.77 | 13.97 | 14.52 | 12.79 | 13.97 | 14.52 | 12.89 | 14.04 | 14.52 |
| 576 | 8.12 | 8.88 | 9.24 | 8.13 | 8.90 | 9.24 | 8.21 | 8.94 | 9.24 |
| 648 | 8.66 | 9.48 | 9.85 | 8.69 | 9.49 | 9.85 | 8.75 | 9.53 | 9.85 |
| 720 | 2.63 | 2.88 | 2.99 | 2.63 | 2.87 | 2.99 | 2.66 | 2.89 | 2.99 |

### 2.5. Potential Sea-Level Rise with Climate Change Scenario

Generally, sea-level rise is caused by the thermal expansion of seawater and the increase in the total volume of seawater due to the melting of glaciers. However, it is difficult to accurately express the dynamics of glaciers due to climate change due to the lack of understanding of the dynamics of seawater and glaciers. In addition, the estimation of the thermal expansion of seawater also contains a great deal of uncertainty. For these reasons, the rate of sea-level rise calculated based on glacier melting and thermal expansion appears to be lower than the rising measured rate [48].

One of the more comprehensive and advanced methods for predicting sea-level rise is the Atmosphere–Ocean coupled General Circulation Model, which combines the Atmosphere General Circulation Model and the Ocean General Circulation Model to calculate the potential sea-level height. This method is based on the thermal expansion of seawater and changes in seawater density obtained from the atmospheric-ocean combination model, so global- and regional-scale sea-level rise can be estimated [49,50].

In this study, the increasing rate of sea-level rise for three target periods of the South Korean coast (East Sea, South Sea, and West Sea) calculated through the Atmosphere-Ocean Circulation Model was applied to determine the potential future sea-level rise and used as input data for the SWMM model (Table 7). These increasing rates were provided from the technical report "Environmental Change Prediction of Natural Disaster and Design Criteria of the Measures for Natural Disaster Prevention and Control Under Climate Change" published by the Korean National Emergency Management Agency (2011) [47]. Notably, the SWMM model can input external flows to nodes and thereby consider the backwater effects such as sea level. Nine outlet points (see Figure 4) were designed to consider the sea-level effect. The sea-level data observed at the tidal monitoring station were used as input and the increasing rates (listed in Table 7) of each target period were added to the current sea-level condition and applied as future sea-level data.

**Table 7.** Predicted potential sea-level rise rate for the oceans around South Korea. Changwon City is located near the south sea (grey) [47].

|  | 2011–2040 (Target I) | 2041–2070 (Target II) | 2071–2100 (Target III) |
|---|---|---|---|
| East sea | 8 cm | 18 cm | 28 cm |
| **South sea** | **10 cm** | **20 cm** | **30 cm** |
| West sea | 12 cm | 24 cm | 32 cm |

## 3. Results

### 3.1. Inundation Analysis Results with Current Climate and Sea-Level Conditions

In this study, an inundation simulation was performed using the SWMM model under the current scenario (i.e., current probable precipitation and sea-level conditions). The scenarios include a total of twelve probable precipitation cases with the frequency of 10, 30, and 50 years and the duration of 1, 3, 6, and 12 h. In this study, to analyze the inundation results from the model, the basin was set as a grid of 15 m × 15 m, and the cells containing the value of inundation depth were investigated. Cells with an inundation depth of 30 cm or less were excluded from the analysis for efficient investigation.

Figure 5 represents the inundation depths resulting from the SWMM model with different probable precipitation events. In the figures, each bar and line indicate the probability density function (PDF) and cumulative distribution function (CDF) of different precipitation frequencies (i.e., 10, 30, and 50 years). According to the different precipitation frequencies, the difference in inundation depths was more pronounced for the 12 h duration than for 1–6 h. In the case of the precipitation scenarios with a 12 h duration, it was found that the 10-year frequency of precipitation had a significant effect on the occurrence of inundation depths of 0.6 m or less, whereas the 50-year frequency of the precipitation pattern mainly contributed to the more considerable inundation depths of 1.0 m or more.

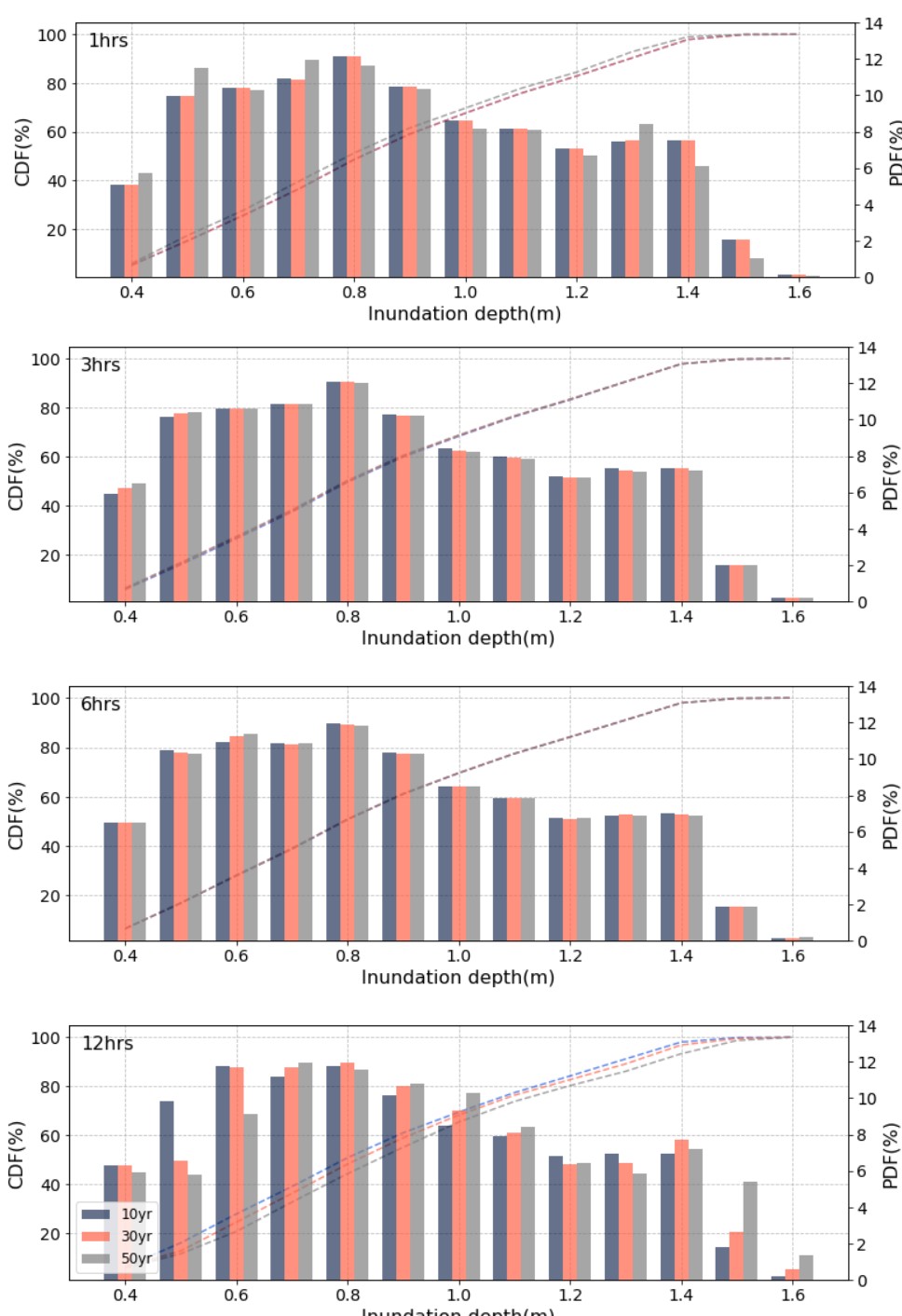

**Figure 5.** PDF and CDF of simulated inundation depth for a current condition.

### 3.2. Inundation Analysis Results with Future Climate Change Scenarios

Future inundation damage was analyzed from the SWMM model using potential precipitation and sea-level rise for three target periods (i.e., target periods I, II, and III). Figures 6–8 represent the analysis results of inundation depth impacted by future precipitation and sea-level rise in urban areas of Changwon City. In the case of target period I, it indicates a pattern similar to the simulation result of the inundation depth by the current condition represented in Figure 5. The significant difference with the present scenario is that the area with a high inundation depth (i.e., >1.4 m) is projected to be more than double the current conditions for all precipitation frequencies and durations. For target periods II and III, the differences in depths increased further.

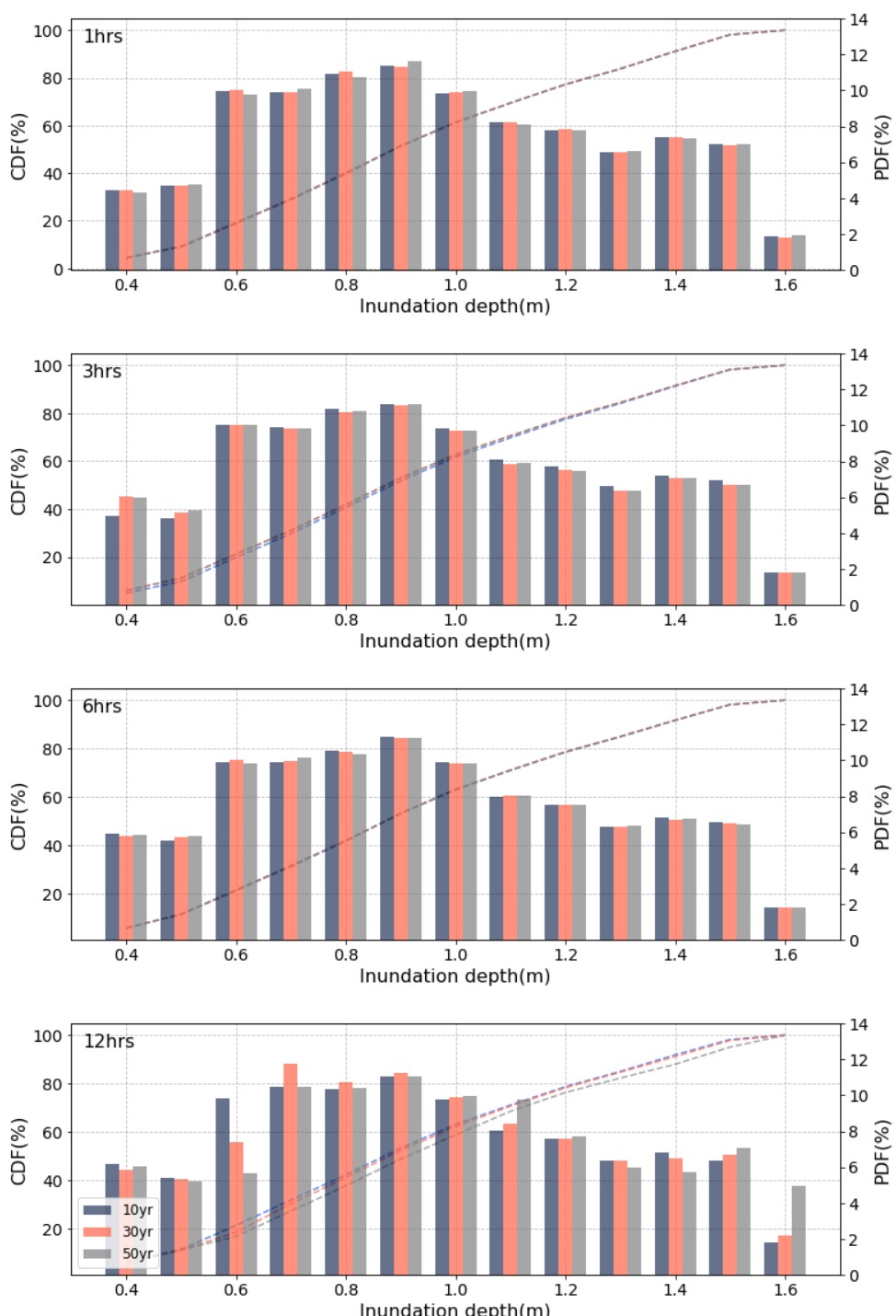

**Figure 6.** PDF and CDF of simulated inundation depth for target period I (2010–2040).

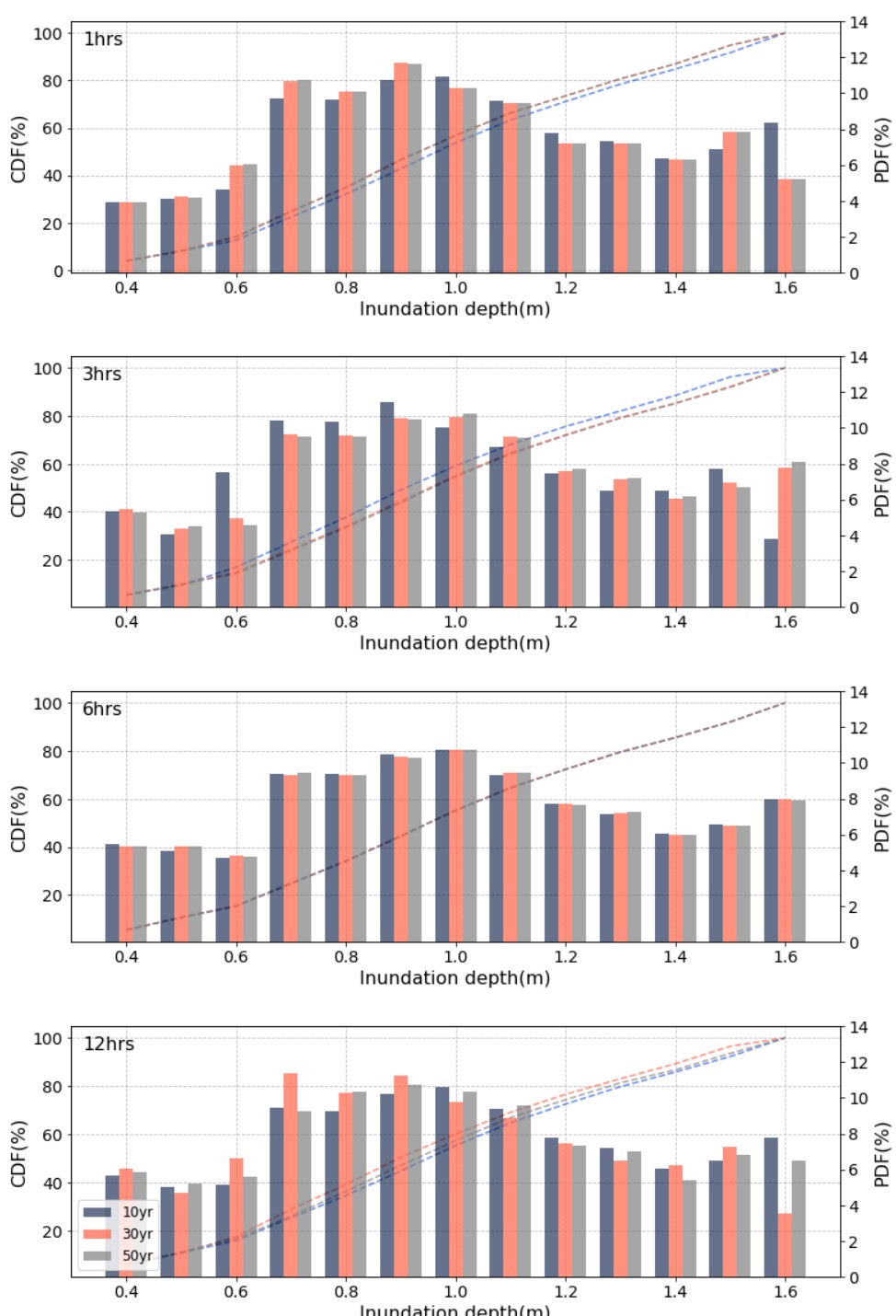

**Figure 7.** PDF and CDF of simulated inundation depth for target period II (2040–2070).

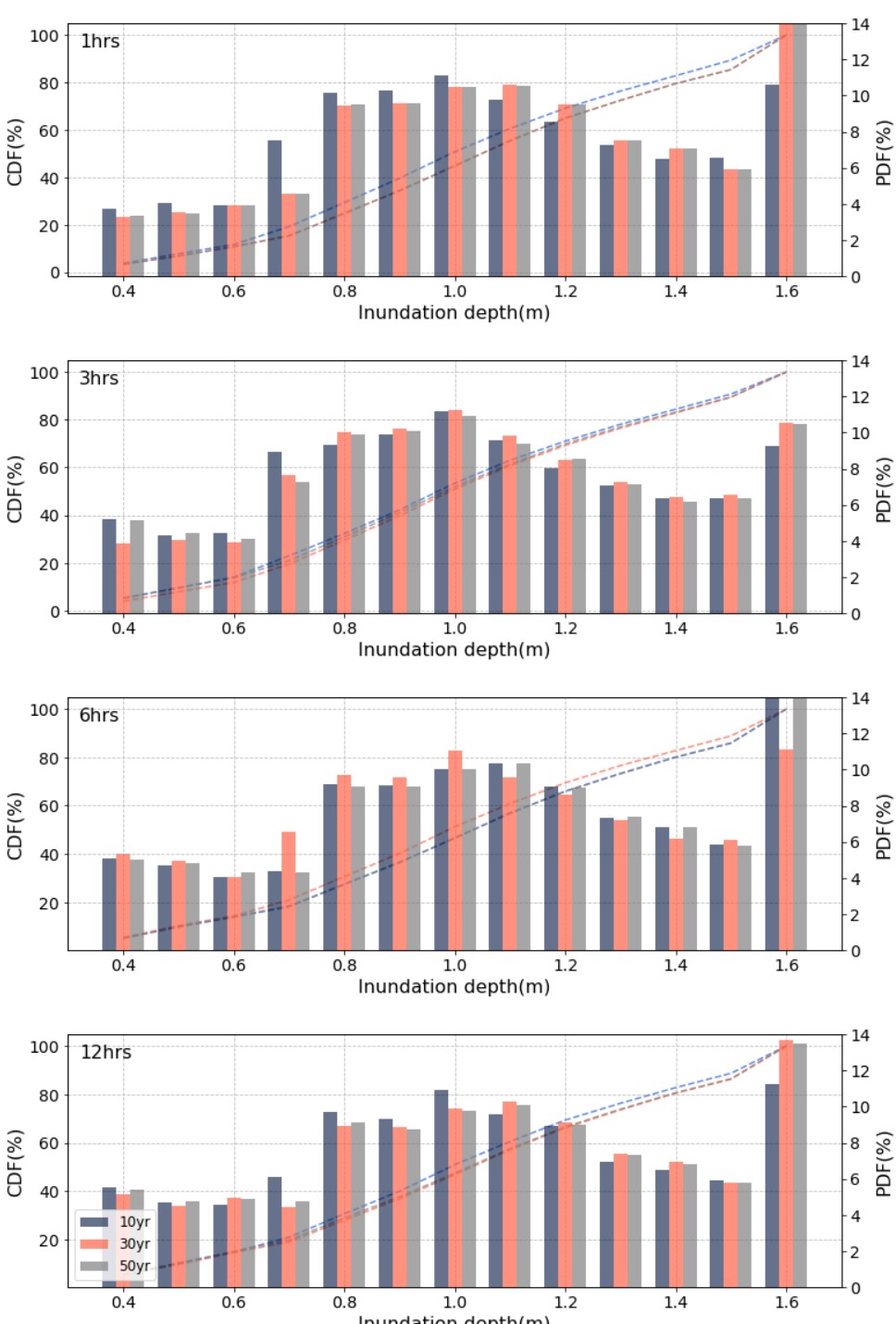

**Figure 8.** PDF and CDF of simulated inundation depth for target period III (2070–2100).

A typical result of future inundation analysis is that the proportion of cells with a higher inundation depth increases as the target period increases. For example, the proportion of cells in which the inundation depth is 1.4 m or more ranges from 1 to 5% in the current or target period I scenario (i.e., 2010–2040), but in target periods II and III (i.e., 2040–2100), it ranges from 6 to 14% of the total flooded area in the city. Moreover, in target period III, 50% of the total flooded area has the potential to generate an inundation depth of 1.0 m or more. Based on the analysis results, assuming that precipitation and sea-level rise are in accordance with the climate change scenario, it seems challenging to mitigate future flood damage with the current sewer pipeline system in Changwon City. In addition,

it can be expected that severe flood damage will occur due to a larger inundation area than the present and a higher depth of inundation.

### 3.3. Potential Flooding Damage under Future Climate Change Scenarios

Figure 9 represents the potential inundation area under future climate change scenarios by comparing it with the current climate scenario of precipitation and sea-level rise. The potential vulnerability to flooding damage in Changwon City increases as the period increases. The rate of the flooded area increased as the precipitation duration decreased. For example, for target period III (i.e., 2070–2100), a modeling result with a duration of 1 h and a frequency of 50 years had the potential to increase the inundation area by approximately 16.2% compared to the current conditions. In contrast, an increase of 6.2% was shown for a duration of 12 h and a frequency of 50 years. For four different durations, the range of the potential increase rate in the inundation area in target period I (i.e., 2010–2040) was from 2.6% to 7.0%, from 3.4% to 10.3% in target period II (i.e., 2040–2070), and from 6.2% to 16.2% in target period III (i.e., 2070–2100).

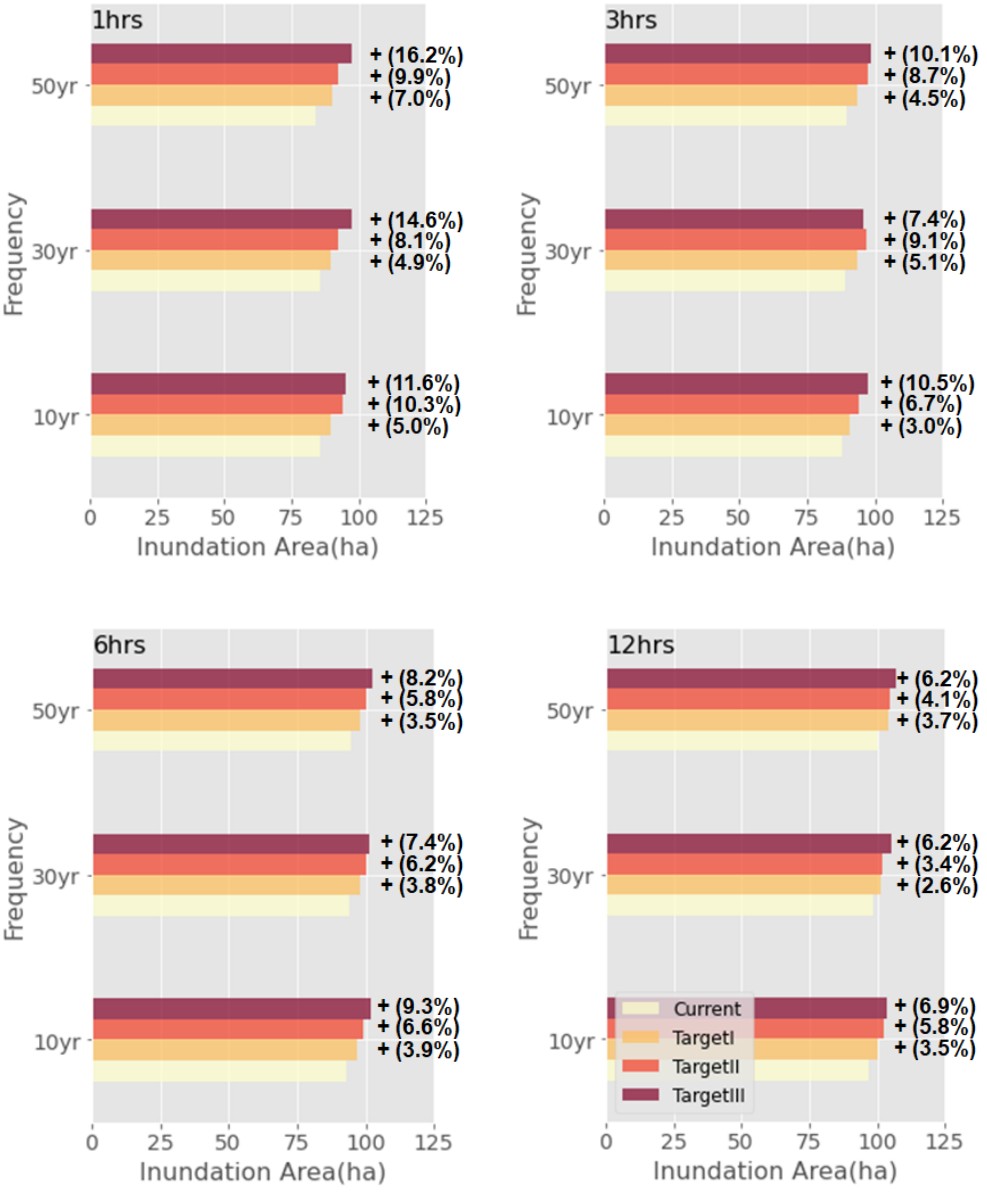

**Figure 9.** Potential inundation area under climate change scenario. Each number in the parentheses indicates an increasing rate compared to the current condition.

As in the analysis of the inundation depth results in Section 3.2, this study indicates that if the amount of precipitation and sea-level rise due to climate change both increase following current trends, the potential inundation area may also increase by up to 16% in the future. In addition, we emphasize the fact that coastal urban areas, which are affected by both precipitation and sea-level rise, may be more vulnerable to flood damage than inland cities.

## 4. Discussions

This study analyzed the inundation damage of coastal urban areas under climate change scenarios. The proposed SWMM model used potential probable precipitation and sea-level rise in the future. Even though the simulated results have shown that both the flooding area and the inundation depth increase as the future target period increases, some limitations should be discussed. The first limitation is the uncertainty regarding the inputs of the model (i.e., pipeline features), which is the main hydraulic input data of the SWMM model. For example, only the main pipelines of the sewage system were considered, and the effect on branch lines was ignored in this study. Although the effect of branch lines on water flow in the sewage system was negligible, it may lead to lower model performance when simulating inundation damage.

The second limitation of this study is uncertainty in climate change scenarios. This is a common limitation of most studies considering climate change scenarios. Since the future precipitation and sea-level data come from GCMs, the uncertainty inherent in the results should be investigated. Therefore, since the prediction results based on climate change scenarios may also have some uncertainties, it is suggested to consider the results to understand the overall potential trends of the increase in the inundation depth or area in the future.

The final weakness of this study is the verification processes of the SWMM model. In this study, as mentioned in Section 2.3, due to the lack of observational data in the area for model verification, the model was verified by comparing it with the actual flooded area caused by typhoon "Sanba" on 16–17 September 2012. Although the verification results confirmed that the modeled flood area was approximately 90% consistent with the actual flooding area, it is suggested that an additional process for model verification should be performed in future studies.

Considering these limitations comprehensively, it is suggested that future studies should consider the process for improving the accuracy of modeling results by taking into account the uncertainty in climate change scenarios, strengthening the model validation process, and improving the reliability of input data.

In addition, in this study, precipitation and sea-level patterns according to the AR4 scenario were used, but it is necessary to consider the latest scenario (i.e., AR6) in the future study [37,51–53]. It is expected that more accurate flood vulnerability simulation and prediction will be possible because the latest global conditions and information such as land management, greenhouse gas fluxes, and terrestrial ecosystems are comprehensively considered in the latest scenarios.

## 5. Conclusions

In this study, the two-dimensional (2D) urban runoff model (i.e., SWMM) was used to simulate the inundation of the coastal metropolitan area of Changwon City according to both the precipitation and sea-level rise in consideration of climate change. The probable precipitation was applied with different precipitation frequencies and durations to evaluate the vulnerability drainage system in the area to flood damage. The frequencies of precipitation divided into the short- (10 years), mid- (30 years), and long- (50 years) term and durations of 1 h–12 h were applied as input data of the model to conduct an inundation analysis. In addition, we used the potential increase rate of precipitation and sea level in Changwon City provided by climate change scenarios to investigate the potential vulnerability to flooding in the future.

As a result of analyzing the future vulnerability to flooding damage in Changwon city by predicting and investigating via the application of climate change scenarios, we determined the local drainage system in the area is dependent on the complex factors of precipitation and sea-level pattern together. It was found that the damage caused by heavy rains and typhoons, etc., which is very vulnerable to flood damage caused by climate change, will be further aggravated.

For the 10-year, 30-year, and 50-year precipitation-frequency scenarios, as the target period increases, both the flooding area and the maximum inundation depth increase compared to the results of the inundation simulation according to the current precipitation and sea-level conditions. The inundation area increased by 2.6–16.2% compared to the current state, and the number of cells with flooded depths of 1 m or more increased. In the case of urban areas with high population density, the flooding damage, with a larger flooded area and a high inundation depth, will be enormous. The vulnerability of coastal cities to flooding damage is expected to increase rapidly due to climate change.

Based on the results of this study, it is essential to consider the comprehensive impact of precipitation and sea-level features on flooding damage in the drainage system of coastal urban areas. Suppose a modeling simulation is performed that considers the actual land use of the watershed and the location or characteristics of the buildings in the metropolitan area. In that case, it is expected that the results of this study will be utilized as valuable primary data for the development of a flood warning system in the future.

Considering the topographical characteristics of South Korea, where three sides of the country are in contact with the ocean and Changwon City, the study area can be expanded to the entire coastal urban area. Future flooding scenarios in the country can be analyzed using future prediction data such as climate change scenarios. It is considered that continuous research is necessary to predict the vulnerability to flooding damage and to come up with a comprehensive strategy.

**Author Contributions:** Conceptualization, H.S.K. and H.H.; methodology, H.H. and D.K.; software, H.H. and D.K.; validation, H.H. and D.K.; formal analysis, H.H.; investigation, H.H.; resources, H.H. and H.S.K.; data curation, H.H. and D.K.; writing—original draft preparation, H.H. and D.K.; writing—review and editing, H.S.K.; visualization, H.H.; supervision, H.S.K.; project administration, H.S.K. All authors have read and agreed to the published version of the manuscript.

**Funding:** This research received no external funding.

**Institutional Review Board Statement:** Not applicable.

**Informed Consent Statement:** Not applicable.

**Data Availability Statement:** Not applicable.

**Conflicts of Interest:** The authors declare no conflict of interest.

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
