# Peer review of "Inundation Analysis of Coastal Urban Area under Climate Change Scenarios"

_water, doi:10.3390/w14071159_

Round 1

Reviewer 1 Report

The article is well written in English and sufficiently clear to the reader. But the methods are dated. For example, the AR4 scenarios were introduced in 2007 and have AR6 ready to come.

-what is your reason for using AR4?

-Please add a highlight to the manuscript.

-Please try to shift two different sections, one for the "Results" and another for the "Discussion".

Author Response

Thank you for the great comments. The authors carefully revised the manuscript according to your comments. Thank you again. 

Reviewer 2 Report

In the paper “Inundation Analysis of Coastal Urban Area under Climate

Change Scenarios” the authors simulate the future inundations of the coastal metropolitan area of Changwon City taking in consideration precipitation and sea-level rise projections from climate scenarios and investigate the way to reduce inundation damages.

The paper offers a comprehensive introduction on the topic, but the methodological choices need to be further explained since they can influence the study.

Below are some comments and clarifications needed:

Why have the authors used the climate model scenarios AR4 to calculate the future climate change patterns and not a more recent one, AR6 or AR5? This needs to be stated and justified since AR4 is more than 10 years old.

The authors do not state which emission scenarios (e.g. A1B, B1, A2) have been chosen for their calculation, the authors state “The climate change scenario presents predicted weather patterns up to 2100 for global climate characteristics. According to the scenario, by 2100, the global average temperature will increase by about 3.7 degrees”; this is not the case since the increase depends on the emission scenarios chosen.

Is SWMM version 4 the one used in the study, if it is the case, please add it in the text. If a higher version has been used, please update the SWMM reference.

In the method section the calibration and validation of details for the SWMM model are missing.

It would be beneficial for the assessing of the results that the authors include a visual representation of the output grid data for the area of study.

The author should add a paragraph about the limitations of the approach presented.

Relevant missing refences

Choo, Yeon Moon, Deok Jun Jo, Gwan Seon Yun, and Eui Hoon Lee. "A study on the improvement of flood forecasting techniques in urban areas by considering rainfall intensity and duration." Water 11, no. 9 (2019): 1883.

Qi, Wenchao, Chao Ma, Hongshi Xu, Zifan Chen, Kai Zhao, and Hao Han. "A review on applications of urban flood models in flood mitigation strategies." Natural Hazards 108, no. 1 (2021): 31-62.

Author Response

(The authors gave the same response as above.)

Reviewer 3 Report

The manuscript presents an inundation analysis of a coastal urban area in S. Korea using SWMM.

Apart from the precipitation the sea level rise is also taken into account.

However, the authors donot present how this sea- level rise in used as an input to the SWMM model. I think that the authors should include a small paragraph for this. 

Also , in the analysis and the conclusions more emphasis should be given to the performance of the model used and the weaknesses of  such a model used for urban floods.

Author Response

(The authors gave the same response as above.)

Round 2

Reviewer 1 Report

I have a serious concern about your study. The problem is, it seems we want to discuss one of the most important natural hazards with the knowledge of 2019, but we are in 2022, and the methods you used are not sufficient for the 2022 climate change situation, especially after the CORONA virus pandemic. The quality of content needs to be improved.

- Please add a paragraph (in introduction or conclusion) and frankly discuss the potential of your work for the latest GCM scenarios and cite new relevant papers(2021-2022). Also, it is better to update your literature review with the latest paper using climate change scenario like this paper:

 https://doi.org/10.3390/su14052601

Author Response

Thank you for the good comments. The authors revised the manuscript according to your valuable feedback. Please see the attachment. 
